# Dimensionality of locomotor behaviors in developing *C. elegans*

Cera W. Hassinan[1,2☯], Scott C. Sterrett[3☯], Brennan Summy[1], Arnav Khera[1], Angie Wang[1,4], Jihong Bai[1,2,3,5]*

**1** Basic Sciences Division, Fred Hutchinson Cancer Center, Seattle, Washington, United States of America, **2** Molecular and Cellular Biology Program, University of Washington, Seattle, Washington, United States of America, **3** Graduate Program in Neuroscience, University of Washington, Seattle, United States of America, **4** Pomona College, Claremont, California, United States of America, **5** Department of Biochemistry, University of Washington, Seattle, Washington, United States of America

☯ These authors contributed equally to this work.
* jbai@fredhutch.org

**Data Availability Statement:** Data are available at https://doi.org/10.5061/dryad.stqjq2c8p and analysis code at https://github.com/ssterrett/wopodyn.

## Abstract

Adult animals display robust locomotion, yet the timeline and mechanisms of how juvenile animals acquire coordinated movements and how these movements evolve during development are not well understood. Recent advances in quantitative behavioral analyses have paved the way for investigating complex natural behaviors like locomotion. In this study, we tracked the swimming and crawling behaviors of the nematode *Caenorhabditis elegans* from postembryonic development through to adulthood. Our principal component analyses revealed that adult *C. elegans* swimming is low dimensional, suggesting that a small number of distinct postures, or eigenworms, account for most of the variance in the body shapes that constitute swimming behavior. Additionally, we found that crawling behavior in adult *C. elegans* is similarly low dimensional, corroborating previous studies. Further, our analysis revealed that swimming and crawling are distinguishable within the eigenworm space. Remarkably, young L1 larvae are capable of producing the postural shapes for swimming and crawling seen in adults, despite frequent instances of uncoordinated body movements. In contrast, late L1 larvae exhibit robust coordination of locomotion, while many neurons crucial for adult locomotion are still under development. In conclusion, this study establishes a comprehensive quantitative behavioral framework for understanding the neural basis of locomotor development, including distinct gaits such as swimming and crawling in *C. elegans*.

## Author summary

Locomotion is an indispensable component of life, which develops effortlessly through adolescence. Locomotor strategies such as running, swimming, and flying are composed of coordinated motor patterns that repeat at stereotypical frequencies. However, the mechanisms governing the establishment and progression of locomotor behaviors in juvenile animals through to adulthood are not well understood. Here we use the nematode *C.*

**Funding:** This research was supported by NIH grants R01NS109476 and R01NS115974 to JB, and F99NS135767 to CWH, R01DC018789, and the Cancer Center Support Grant P30 CA015704 to Fred Hutchinson Cancer Center. The funders had no role in study design, data collection or analysis, decision to publish, or preparation of the manuscript.

**Competing interests:** The authors have declared that no competing interests exist.

*elegans* well-defined time course of development, fully reconstructed connectome, and repertoire of simple locomotor behaviors to study the development of locomotion. Using recent developments in quantitative behavioral methods, we observed and measured swimming and crawling behaviors in *C. elegans*. We found that swimming and crawling are characterized by rhythmic patterns of distinct sets of body postures, called "eigen-worms." Newly hatched worms are capable of producing adult-like locomotor postures, albeit in unsteady bouts which become more stereotypical across development into adulthood. These improvements in locomotor stability coincide with previously known neuro-developmental milestones early on in post-embryonic development. Our findings contribute to a growing trend towards leveraging quantitative methods to capture the complexity of naturalistic behaviors and are a point of reference for studying developmental programs important for locomotor development.

## Introduction

Locomotion, including behaviors such as running, flying, swimming, and crawling, is vital for animals to navigate their surroundings. These behaviors emerge early in life, enabling animals to interact with and adapt to their environments. For instance, juvenile worker bees can perform in-hive tasks, tadpoles can swim and feed, and precocial animals like horses can mobilize soon after birth [1–3]. However, juvenile animals are often incapable of performing the full repertoire of smooth rhythmic locomotion that they will eventually develop [1–8]. Organized locomotion is subject to developmental regulation, as animals undergo various anatomical changes, requiring a maturing nervous system to continuously accommodate a growing body [9–13]. For instance, early development during the prenatal period in humans lays down the primary structure of the brain but neural networks undergo further refinement to stabilize, adapt, and reshape their control of a growing body to produce structured rhythmic behavior [14–16].

In both vertebrates and invertebrates, the nervous system relies on a series of cell signaling pathways to generate the rich diversity of neurons and glia required to perform complex behaviors [17–21]. Progressing into adulthood, synapse formation and neuronal remodeling are pivotal steps in development where intricate neural circuits will forge necessary connections for a stable behavioral output [22–25]. Significant research efforts have led to an understanding of conserved developmental mechanisms that define neuron specification, migration, and wiring at various life stages. However, how developmental pathways set up neural circuits to produce rhythmic locomotion remains unknown.

Rhythmic locomotion is characterized by repetitive, structured postures that enable continuous coordinated movement [26]. This behavior depends on a number of factors, such as neural processing of sensory information and the assembly of central pattern generators (CPGs) for efficient motor control [27,28]. CPGs, named for their intrinsic ability to generate rhythmic activity of motor neurons, are groups of neurons central to producing rhythmic locomotion [29–32]. Sensory input, although not essential to rhythmic patterning, plays a vital role in generating appropriate motor commands and ensuring accurate locomotion [27]. Gaining clarity on the maturation of animal locomotor behavior could significantly enhance our understanding of the development of complex neural circuits underlying rhythmic locomotion.

To understand how coordinated rhythmic locomotion matures during development, we took advantage of the nematode *Caenorhabditis elegans* due to its compact and well-defined

nervous system, simple anatomy, and minimal yet reliable production of rhythmic locomotor behaviors. Indeed, *C. elegans* development is extensively characterized, including the known lineage of all its cells [33–36]. This detailed knowledge allows investigations to pinpoint neuron function related to behavioral phenotypes, as early as embryonic stages [37]. Prior research efforts have led to a systematic understanding of the worm's wiring diagrams across postembryonic development [13]. These extensive studies reveal that young L1 larvae have 222 neurons and approximately 1,500 synapses, while the adult *C. elegans* nervous system consolidates 302 neurons and around 8,000 synapses. These findings illustrate the drastic remodeling of the nervous system over the course of development [13,38–40].

*C. elegans* perform two fundamental locomotor behaviors in their natural environment–crawling and swimming [41–43]. These movements are facilitated by rhythmic body bends composed of alternating dorsal and ventral muscle contractions along the length of the body [32]. Prior studies on *C. elegans* locomotion have primarily investigated the body bend frequency in adult crawling and swimming [41,44–46]. Furthermore, the crawling behavior of adult animals has been quantitatively described at the postural level, revealing stereotypical sinuous movements [47,48]. Yet, the question of whether crawling and swimming share a common gait remains contested [41,44–46,49–52]. Moreover, the progression of coordinated locomotion across development is yet to be determined.

Here, we have delineated the diverse locomotor postures exhibited by *C. elegans* during post-embryonic stages through to adulthood, with the aim of unraveling the developmental progression of rhythmic locomotion. Our findings quantitatively show that swimming and crawling behaviors are categorically distinct in frequency, postural, and principal component analyses. We have also assessed the coherence and stability of swimming and crawling postures at different developmental stages to identify the time point at which they mature. Interestingly, we find juvenile *C. elegans* can generate coordinated locomotor rhythms, despite their small body size and the ongoing restructuring of the nervous system. However, this ability only manifests later in the L1 larval stage, indicating a critical developmental event that triggers the acquisition of this capability.

## Results

### Adult *C. elegans* swimming is described by mixtures of unique shapes

*C. elegans* swim by propagating dorsal and ventral waves along the length of their bodies in a movement commonly referred to as thrashing. This behavior exemplifies rhythmic locomotion, as adult wildtype *C. elegans* can continuously bend their bodies for hour-long timescales before transitioning into resting states [53]. Previous studies have described the basic kinematics of these swimming movements by focusing on body bend frequency and curvature [41,44]. Building upon this, we used tracking methods to monitor day-1 adult *C. elegans*, thereby extracting their swimming postures. We divided the worm's body into 11 equal segments, allowing us to measure the angle between adjacent segments (Fig 1A). This approach resulted in a breakdown into 10 distinct angles, offering an in-depth view of the posture dynamics during swimming. Through quantifying the worm's swimming curvature over a 10-second period, we confirmed the consistent rhythmic pattern produced by adult *C. elegans* (Fig 1B).

To comprehensively understand the repertoire of swimming postures, we performed principal component analysis on the curvature data of adult swimming worms and found that the continuous body bends can be succinctly described through this approach. This method has previously been used to describe the low dimensional structure of adult crawling [47]. Through this approach, the repertoire of swimming postures can be reconstructed from the linear combination of principal components, or "eigenworms" (Fig 1C and S1 Table). We

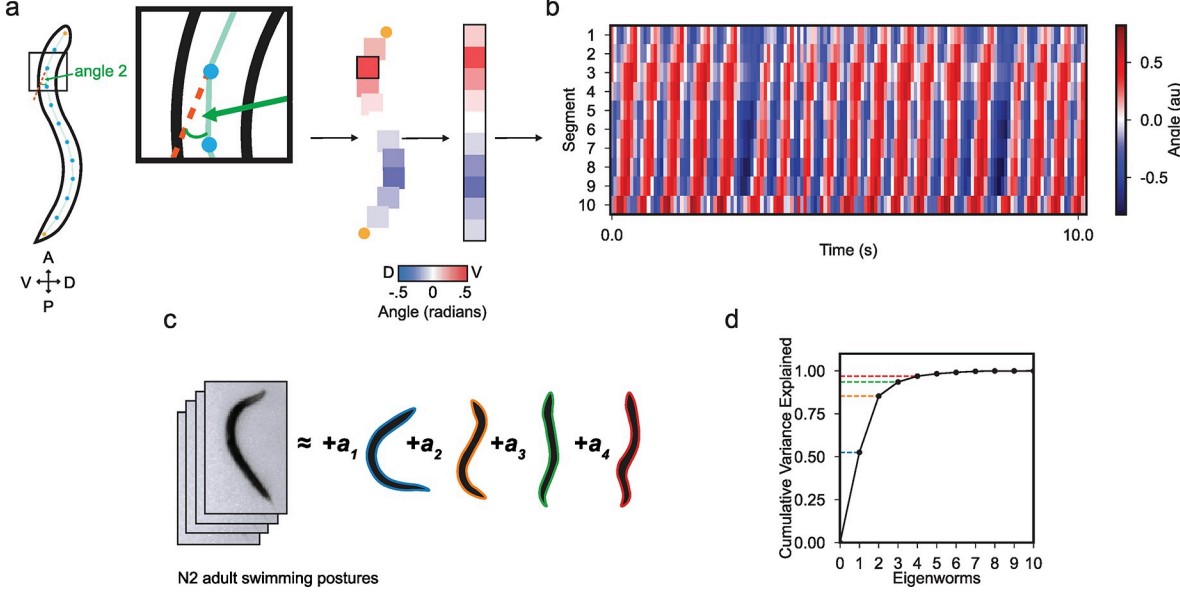

**Fig 1. Adult *C. elegans* swimming is low dimensional.** (a) Schematic showing the conversion of a worm body posture into curvature angles at a single time point. The shape of the worm is divided into 11 segments along the anterior-posterior axis and angles are calculated between adjacent segments. The 10 angles are converted to a colormap with blue-to-red indicating dorsal-to-ventral deflections. (b) Kymograph of body curvature of a day-1 adult worm during 10-seconds of swimming. Body segment number is plotted on the y-axis and time on the x-axis. (c) The variance in swimming postures of adult worms is largely captured by a linear combination of four eigenworms. The first four eigenworm shapes are reconstructed and are outlined in blue, orange, green, and red, respectively. (d) The fraction of the total variance explained in swimming postures as a function of the number of eigenworms used in reconstruction of adult *C. elegans* (n = 43) swimming behavior. Dashed colored lines indicate the cumulative variance explained by the first four eigenworms at 53%, 85%, 94%, and 97%, respectively.

found that a minimum of four eigenworms can explain 97% of the variance in adult *C. elegans* swimming postures. Furthermore, the first two eigenworms account for a significant portion of the variance in swimming postures, explaining 85% of the total variation (Fig 1D). These data indicate that swimming behavior in adult *C. elegans* has low dimensionality and that a few eigenworms can describe common postures the animals make when swimming.

## Distinct eigenworms describe crawling and swimming in adult *C. elegans*

After identifying that swimming behavior in adult worms is low dimensional, we next investigated the structure of these eigenworms and their relationship to crawling. Eigenworms represent correlations between body segment angles, so their shape represents common postures of the worm during locomotion. We found that the first two swimming eigenworms reflect the stereotypical "C-shape" seen in swimming and a sinous, "S-shape", respectively (Fig 2A). Being able to describe the posture of a swimming worm using a linear combination of the eigenworms, we next investigated how the eigenworm amplitudes vary during swimming. A swimming adult worm propels itself through alternating dorsal and ventral muscle contractions generating rhythmic sinuous waves. We found that these oscillations are represented as cycles in the ring-like structure of the first two eigenworm amplitudes during swimming (Fig 2B and S1 File). This ring structure indicates that combinations of the first two eigenworm amplitudes represent the phase of the coordinated dorsal-ventral body oscillations of a swimming worm. The presence of a ring-like structure in the distribution of the first two eigenworm amplitudes is qualitatively similar to that found in *C. elegans* crawling [47]. However, we discovered that the principal two eigenworms found in swimming locomotion are distinct

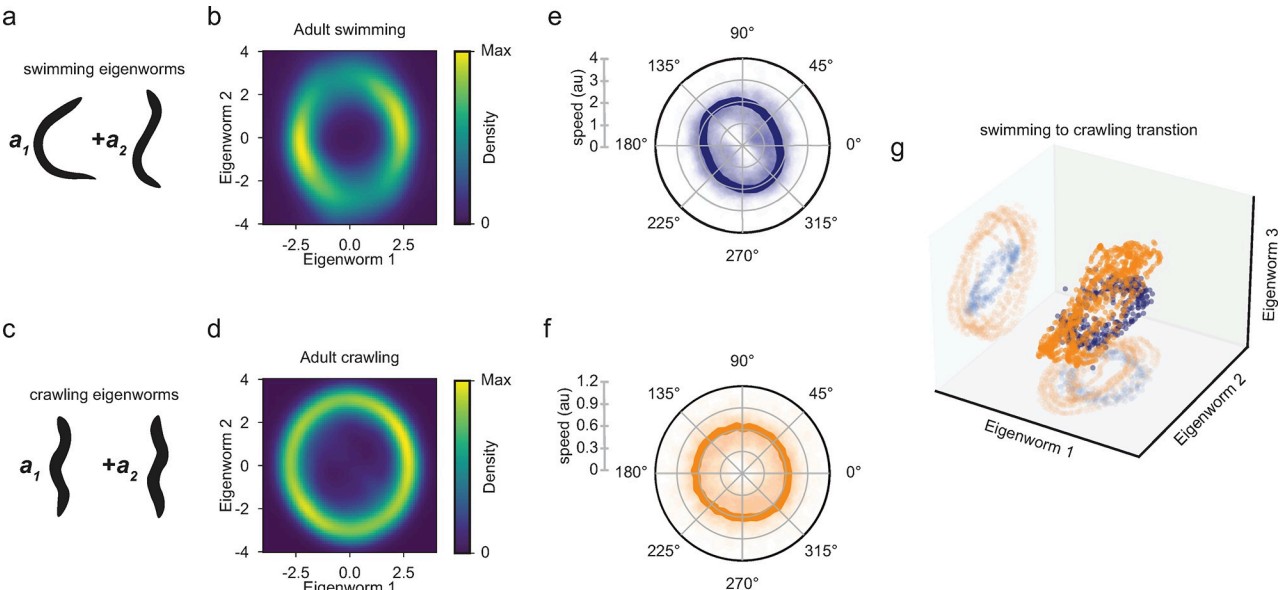

**Fig 2. *C. elegans* adult swimming and crawling are distinct gaits.** (a) Schematics of the first two principal eigenworms from adult swimming as shown in Fig 1C. (b) Swimming eigenworm amplitude distributions show a stereotyped ring structure which captures the coordination of swimming eigenworms one and two to produce swimming locomotion in adult *C. elegans* (n = 43). (c) Schematics of the first two principal eigenworms of adult crawling behavior. (d) Crawling eigenworm amplitude distributions show a stereotyped ring structure of coordination between crawling eigenworms one and two in adult *C. elegans* (n = 38). (e and f) Polar plots of eigenworm amplitude (b and d) speeds as a function of phase in the ring. Speed data across all animals are plotted as scatter points, and the mean is overlayed. (e) In swimming, speed is bimodal and is slowest when in the "C" shape, or eigenworm one, whereas in crawling (f) the speed is constant along the ring. (g) 3D scatter plot of the first three eigenworm amplitudes from a representative worm tracked during a swimming-to-crawling transition. The ring structure of swimming (blue) is distinct from the ring structure associated with crawling (orange).

from the principal two eigenworms discovered in crawling locomotion (Fig 2A and 2C). This demonstrates that crawling and swimming locomotion produce categorically different body postures. Our crawling analyses confirm previous findings that crawling locomotion produces two sinuous eigenworms which form a ring-like structure in the first two crawling eigenworm amplitudes (Fig 2D and S2 File) [47].

When comparing swimming and crawling rings, we also found that the swimming ring contains two peaks in density, whereas the crawling ring is essentially uniform in density (Fig 2B and 2D). We hypothesized that this peaked structure in swimming arises due to differences in the speed of oscillations in eigenworm space; if a worm moves slower during a phase of the swimming cycle, we will observe more frames in this posture and vice versa. To test this hypothesis, we calculated the speed of trajectories in eigenworm space and plotted the average speed as a function of phase in the ring. These speed analyses confirm our hypothesis that during swimming locomotion the worm moves slower when eigenworm one is largest in amplitude and faster when eigenworm two is largest in amplitude (Fig 2E and 2F). This indicates that swimming worms spend more time in the stereotypical C-shape and less in the sinuous shape. However, crawling locomotion shows a uniform speed throughout the entire cycle indicating that crawling behavior is a constant propagating wave.

To further demonstrate our finding that swimming and crawling locomotion produce disparate postures, we performed a gait transition assay, where we track an individual adult worm as it transitions from swimming to crawling. When we performed principal component analysis on these postural data, we found that the first four eigenworms are composed of the principal two eigenworms found in swimming and the principal two eigenworms found in

crawling. When we plot the first three eigenworm amplitudes from a transition assay, we observed that swimming and crawling locomotion trace out separate rings. This finding shows that swimming and crawling rings exist in separate regions of posture space (Figs 2G and S1). These analyses highlight that swimming and crawling locomotion use distinct postures.

## Young L1 *C. elegans* generate but cannot reliably maintain swimming rhythmicity

*C. elegans*, like all animals, undergo dramatic size changes throughout their development. They grow from approximately 0.2 mm in length immediately after hatching to roughly 1 mm as day-1 adults (Fig 3H). Alongside these physical transformations, postembryonic development in *C. elegans* also involves substantial remodeling of the nervous system. To determine the impact of development on rhythmic swimming locomotion in *C. elegans*, we studied the low-dimensional structure of body postures during development. We carried out swimming assays at various developmental larval stages–young L1, late L1, L2, L3, and L4. Interestingly, we found that the first four eigenworms in swimming are highly similar across all developmental stages (Fig 3A). This finding suggests that, even at early post-embryonic developmental stages, worms are capable of executing swimming postures that closely resemble those of fully grown adults.

Furthermore, when we examined eigenworm amplitudes across development, we found that young L1 swimming has a stereotypical ring-like structure, indicating adult-like coordination. However, there is a notable increase in density at the center of the ring when compared to later developmental stages (Fig 3C), corresponding with irregular coordination of postures (S3 File). Thus, while all developmental stages are capable of producing adult-like swimming postures, young L1 animals are unable to sustain rhythmic locomotion for the duration of the recording. This result suggests that the ability to sustain rhythmic locomotion emerges at the late L1 stage and is retained throughout the developmental progression into adulthood (Figs 2B and 3D–3G).

To quantify these differences, we calculated the dimensionality of the swimming behavior across different developmental stages using a metric known as the participation ratio (PR). The PR is a continuous number representing the dimensions required to describe approximately 80% to 90% of the variance in a dataset (see Methods and S1 Table) [54]. We found that young L1 swimming has a mean PR of 3.25, which is significantly higher than all other developmental stages with means of 2.63, 2.50, 2.57, 2.54, 2.49 for late L1, L2, L3, L4, adult respectively (Fig 3B). This indicates a higher dimensionality at this stage, corresponding to a less structured locomotor repertoire. In summary, the swimming behavior of young L1 animals, despite its higher-dimensional structure, exhibits the same principal four eigenworms observed in adults. This suggests that young L1 worms are capable of performing, but unable to sustain, adult-like rhythmic swimming behaviors. Furthermore, postural swimming patterns become more organized during development, indicating potential developmental mechanisms associated with this stabilization of swimming behavior.

## Young L1 *C. elegans* have immature crawling postures and locomotor patterns

In light of our finding that adult swimming and crawling constitute distinct gaits, it was crucial to examine if these locomotor behaviors follow similar developmental timelines. Thus, we performed behavioral crawling assays across various developmental stages and conducted principal component analysis on the posture data. Interestingly, we found that young L1 eigenworm shapes in crawling are different from all other developmental stages (Fig 4A). All deviations

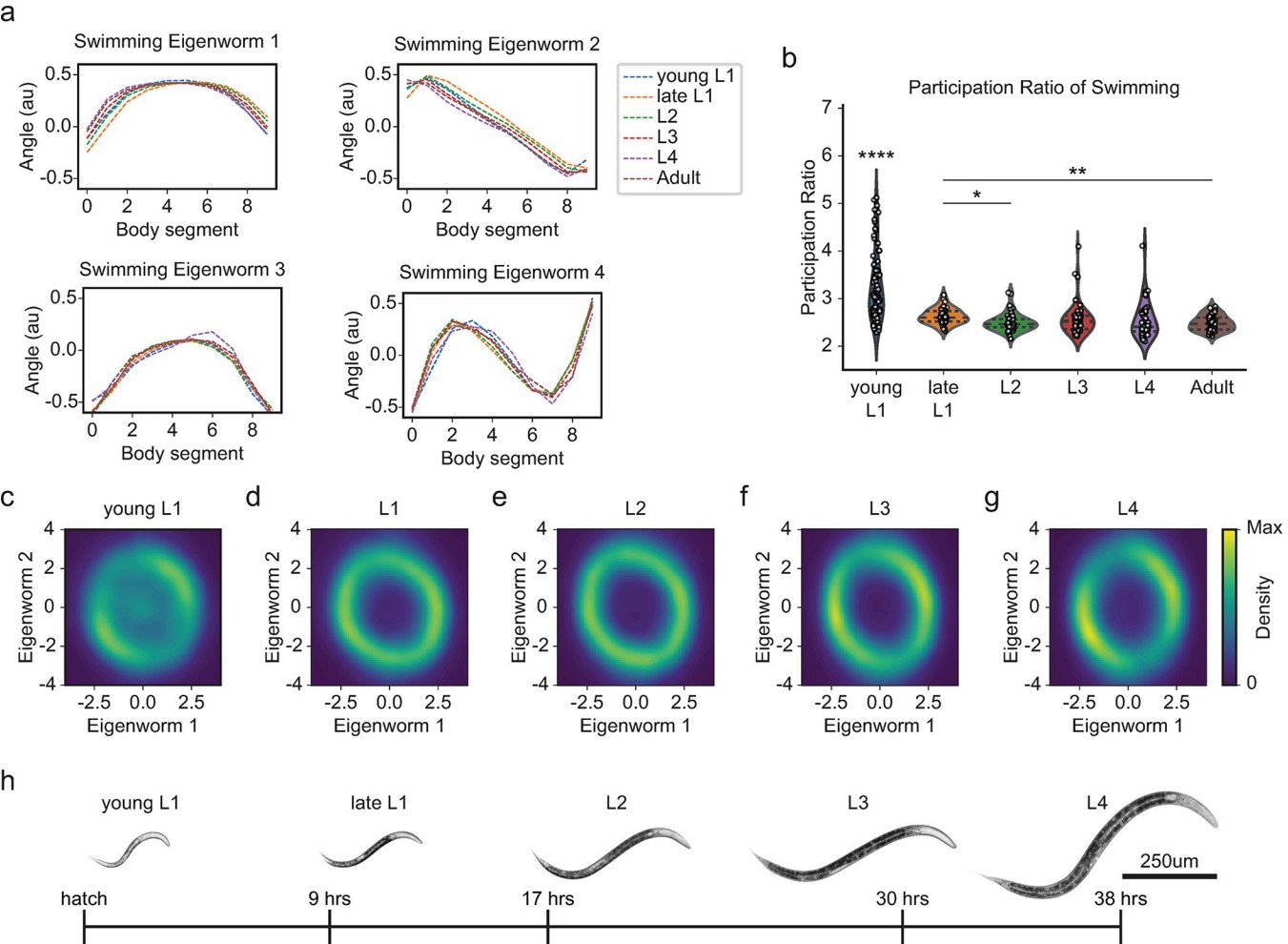

**Fig 3. Rhythmic swimming is present at birth and matures throughout development.** (a) Swimming eigenworms 1–4 across developmental stages: young L1 (blue), late L1 (orange), L2 (green), L3 (red), L4 (purple), adult (brown). (b) Participation ratios (PRs) representing the dimensionality for each swimming tracking session of young L1 (n = 86), late L1 (n = 40), L2 (n = 47), L3 (n = 48), L4 (n = 39), and adult (n = 43) *C. elegans*. Young L1 and adult *C. elegans* swimming PRs show a significant difference in means (p = 9.07e-08, t-test). (c-g) Swimming locomotion represented by eigenworm one and two amplitude distributions across developmental stages: young L1 (c), late L1 (d), L2 (e), L3 (f), and L4 (g) demonstrate coordination of these eigenworms is present across development, however young L1 worms also produce uncoordinated postures not represented by the first two eigenworms. (h) The developmental stages, young L1, late L1, L2, L3, L4 of N2 *C. elegans* recorded in this study. Dashed lines in (b) represent means and interquartile range. Statistical significance in (b) was determined using Bonferroni adjusted alpha levels of 0.03 (0.05/15). Young L1 PRs showed ****p statistical significance compared to all other stages. Significance: *p<0.0033, **p<0.00067, ****p<0.0000067.

were observed in the posterior half of the worm, indicating that the posterior region lags behind the anterior half in forming mature crawling postures. While these body postures are distinct, the overall shape still resembles the sinuous postures necessary for the assembly of coordinated crawling behaviors in adult animals. By the late L1 stage, these eigenworm shapes had transitioned into the adult form, suggesting that an immature locomotor circuit in young L1 worms must undergo critical changes in order to support crawling poses across the full length of a worm.

Next, we examined whether young L1 worms could assemble rhythmic crawling patterns like those observed in adults. This was accomplished by measuring the distribution of the first two eigenworm amplitudes. Surprisingly, we found that young L1 crawling worms exhibit a complete loss of ring structure in the eigenworm amplitude distributions (Fig 4C). These

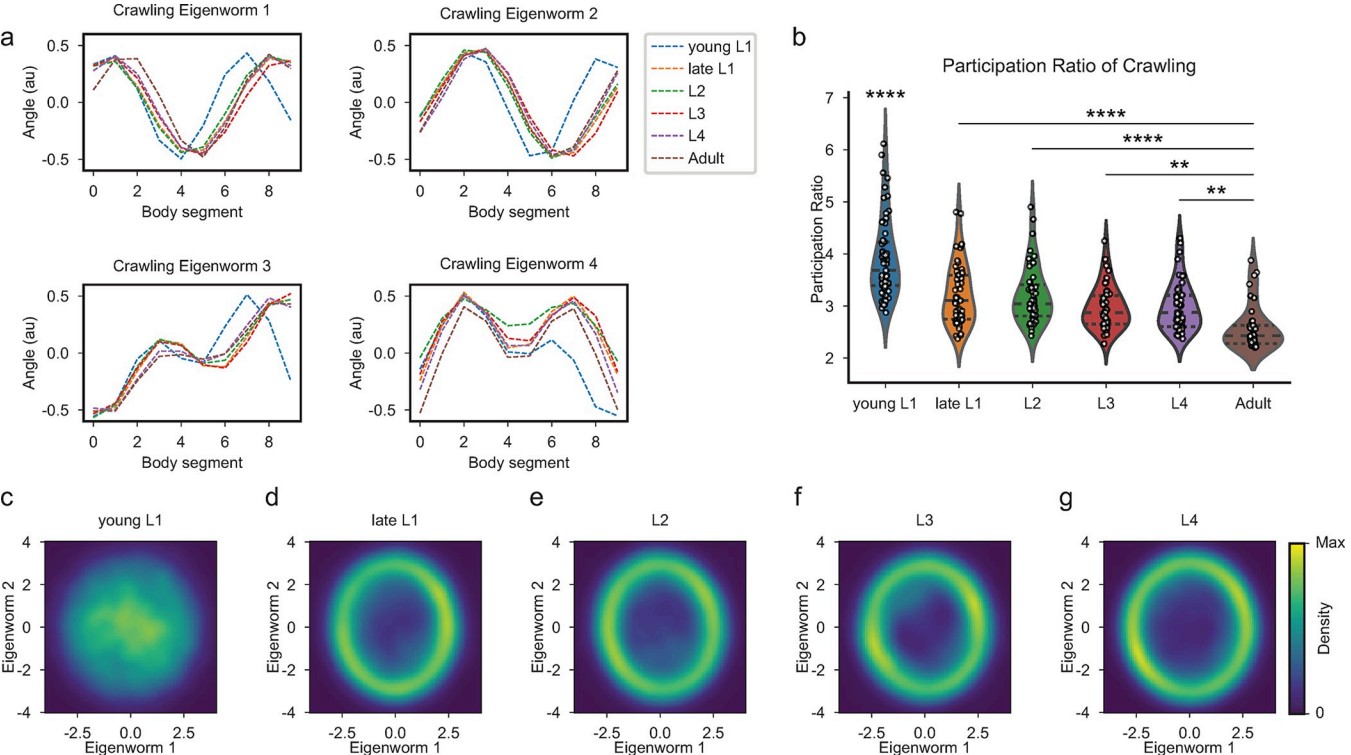

**Fig 4. *C. elegans* crawling postures are disrupted at birth and stabilize by late larval 1 stage.** (a) Crawling eigenworms 1–4 are identical across developmental stages: late L1 (orange), L2 (green), L3 (red), L4 (purple), adult (brown), except for posterior deviations along the body at the young L1 stage (blue). (b) Participation ratios (PRs) for each crawling tracking session of young L1 (n = 61), late L1 (n = 51), L2 (n = 51), L3 (n = 43), L4 (n = 47), and adult (n = 38) *C. elegans*. Young L1 and adult *C. elegans* crawling PRs show a significant difference in means (p = 1.87e-14, Welch's t-test). (c-g) Crawling eigenworm one and two amplitude distributions across developmental stages: young L1 (c), late L1 (d), L2 (e), L3 (f), and L4 (g) demonstrate the coordination of these eigenworms is not present at the young L1 stage but develops by the late L1 stage. Dashed lines in (b) represent means and interquartile range. Statistical significance in (b) was determined using Bonferroni adjusted alpha levels of 0.03 (0.05/15). Young L1 PRs showed ****p statistical significance compared to all other stages. Significance: **p<0.00067, ****p<0.0000067.

findings reveal that young L1 animals are unable to produce the synchronous body waves that are characteristic of mature crawling (S4 File). The distinct eigenworms and lack of organized crawling postures in young L1 animals suggest the existence of distinct locomotor circuits for crawling and swimming at this stage. However, similar to swimming, the coordination of crawling postures has become clearly defined by the late L1 stage and preserved throughout subsequent developmental stages (Fig 4D–4G). These findings suggest that both crawling and swimming behavior mature during the L1 developmental period. To further understand the variability of postures in crawling data across development, we quantified the dimensionality of crawling behaviors using PR. We found that young L1 crawling has a mean PR of 3.90, significantly higher than all other developmental stages with means of 3.22, 3.21, 2.98, 2.99, 2.58 for late L1, L2, L3, L4, adult respectively (Fig 4B). Our results show that young L1 animals exhibit considerable uncoordinated crawling, but as development progresses, locomotor patterns become more stable by the late L1 stage, highlighting the significant maturation of locomotor circuits that occurs during this critical developmental period.

## Discussion

*C. elegans* perform two fundamental rhythmic locomotor strategies dependent on their surroundings–swimming and crawling. Building on advances in pose-estimation and behavioral

analysis [55], here we demonstrate that *C. elegans* rely on mixtures of locomotor postures unique to swimming and crawling. These postures are present in swimming young L1 larvae and are maintained into adulthood. However, in crawling young L1 larvae, eigenworms are immature and develop by the late L1 stage. Although these basic postures are present in young L1 larvae, they struggle to sustain coordination. Development leads to a stabilized coordination of locomotor patterns, allowing for the production of robust rhythmic locomotion as early as the late L1 stage. We also extend our analysis to identify features of gait transitions by distinguishing swimming and crawling behaviors in eigenworm space. Together, our findings establish a quantitative basis for unraveling the link between locomotion and the significant remodeling of sensorimotor circuits in a developing nematode.

## An innate nature of locomotor patterning

The *C. elegans* locomotor circuit is composed of sensory neurons and interneurons that orchestrate excitatory and inhibitory motor neuron output [32,42,52,56–59]. Recent research has shown that L1 animals, which have an incomplete motor circuit lacking a subset of motor neurons, depend on extrasynaptic signaling for the facilitation of "adult-like" bending waves [13,60]. In a related fashion, we found that young L1 animals have characteristics of swimming and crawling seen in adult animals. Thus, variable circuits can produce similar behaviors making this a salient example of circuit degeneracy, where diverse neural structures can yield similar functional outcomes. In *C. elegans* natural habitat of diverse decaying plant materials, they are required to navigate soil and liquid after hatching [61]. We speculate that young L1 *C. elegans* survival necessitates the ability to generate rudimentary locomotor postures from immature neural circuitry.

Our results show that crawling and swimming behaviors are highly stereotyped, starting from the late L1 larval stage and continuing into adulthood. This stable pattern of motor execution might be attributed, in part, to the stereotyped nature of neuromuscular connections. Prior connectomics studies across various species have shown a pattern–while other types of neuron connections exhibit high variability, motor neuron connectivity typically maintains a strict stereotypy during development. This dichotomy suggests that the stability and fidelity of motor execution may be reinforced by the lower variability inherent in motor neurons' connections, in contrast with the higher variability observed in the output connections from modulatory neurons [13]. However, despite this stability, the nematode locomotor circuit undergoes significant remodeling at early development. Specifically, within the ventral nerve cord motor neurons, a total of 53 neurons are added during the late L1 stage, resulting in substantial rewiring [42,56, 62–65]. Despite this intensive period of neuronal network restructuring and rapid growth, *C. elegans* remarkably manage to generate coordinated rhythmic locomotion. Moreover, previous research has shown that *C. elegans* locomotion tolerates the inactivation of key neurons in the locomotor circuit [57,66–71]. Taken together, these observations underscore the resilience and robust nature of locomotion in *C. elegans*, even amidst significant developmental changes.

## The robustness of locomotor coordination is acquired during development

It is generally thought that networks of neurons need to be appropriately assembled into functional circuits to produce smooth motor outputs. For instance, CPG neurons produce the basic rhythm of locomotion [29–31,72]. However, without sensory feedback, CPGs alone cannot generate smooth movements despite their intrinsic rhythmic activity [31,73–75]. We demonstrate that young L1 animals fail to maintain an organized crawling or swimming rhythm. We hypothesize that immature neural circuits in young L1 worms cannot fully integrate

sensorimotor feedback into rhythmic circuits required for the coordination observed in adult animals. In fact, during the late L1 developmental stage, the nervous system undergoes significant changes with the addition of various neurons and substantial rewiring, potentially compensating for early uncoordinated locomotion [63,76–79].

Notably, we observe young L1 animals exhibit deviations in crawling postures, specifically in the posterior half of the body and display considerable uncoordinated crawling. We hypothesize that swimming and crawling behaviors are potentially the output of discrete circuits at the young L1 larvae stage. *C. elegans* sense surrounding physical forces prompting distinct crawling and swimming locomotor strategies. It will be informative to investigate the role of neurons born later in development. For instance, post-embryonic sensory neurons, AVM and PVM, emerge during the L1 stage and are thought to play a role in sensing gentle touch in the anterior and posterior regions of the body important for forward and backward locomotion, respectively [76,80–83]. It is possible that crawling is limited at this stage due to the absence of sensory neurons that modulate CPGs. Additionally, stable locomotor behavior could also be subject to muscle development [84, 85]. Our findings that there is a general decrease in dimensionality across development into adulthood, where locomotor behavior is most stereotypical, will be a useful benchmark for future studies of developmental mechanisms.

## Swimming and crawling use different gaits

We find that swimming and crawling gaits are distinct in both their postures and rhythms. Prior studies in *C. elegans* have shown that dopamine and serotonin play critical roles in the swimming-to-crawling and crawling-to-swimming transitions, suggesting that conserved mechanisms are used for gait transitions across animal species [44]. Yet, the debate continues over whether crawling and swimming gaits represent the output of distinct or shared neural circuits [41,45,46,49,50,59]. Our gait transition analysis in adult worms demonstrates that swimming and crawling behaviors are posturally distinct as they trace out separate rings in eigenworm space. Additionally, the analysis of angular speed shows variability in the rhythm underlying swimming and crawling, with a more uniform distribution in crawling. Our findings raise the question of whether animals can transition between gaits across development and if gait selection is primarily a response to physical sensation.

## The power of quantitative behavioral analysis

Eigenworm analyses provide a quantitative and reproducible framework which enhances the reliability and comparability of behavioral data across studies. Indeed, when comparing our eigenworm analysis of adult crawling with previously published work [47], we confirmed many of their key findings: we found that four eigenworms describe over 95% of the variance, that these four eigenworms have the same shape, and that the first two eigenworms trace out a ring structure. A thorough quantitative understanding of behavior is an important first step before subsequent studies of neural circuit function. Furthermore, the precise temporal nature of these behavioral analyses are well suited for neural circuit investigations where variation at the sub-second timescale is important [86]. For example, recent research using eigendecomposition of embryonic postures, so called "eigen-embryos", shows that motor behaviors mature embryonically and are modulated by specific neurodevelopmental processes such as RIS neuron activation [37]. Longer timescale measurements like undulation frequency, distance traveled, and speed, are often incapable of matching these fast neural timescales. It is from this point of behavioral understanding that studies of the neural mechanisms can be launched.

Our results have not only advanced the understanding of locomotor development and gait transitions in *C. elegans* but have also laid a foundational framework for future investigations

into the modulatory mechanisms that drive the establishment, maintenance, and flexibility of rhythmic locomotion during nervous system development.

## Materials and methods

### Preparation of worms

Wild type hermaphrodite *C. elegans* (N2 Bristol) worms from the CGC (Minneapolis, MN, USA) were used for all assays. The worms were maintained at 15˚C on 60mm NGM agarose plates with Escherichia coli OP50 lawns as food. To obtain worms at later stages of development, synchronization procedures were carried out in which 20 gravid hermaphrodites were placed on seeded NGM plates. After one hour, all worms were subsequently removed, leaving only eggs on the plates which were immediately put into a 20˚ incubator. After 21 hours of incubator growth time for late L1s, 29 hours for L2s, 42 hours for L3s, 50 hours for L4s, and 65 hours for adult worms, the synchronized animals were subjected to assays. Young L1 recently hatched animals were collected by transferring 50 gravid hermaphrodites to a seeded plate 24 hours before experimental assay. On the day of the experiment, the plate with gravid hermaphrodites and laid eggs was washed with M9 buffer until all animals and bacteria were removed leaving only eggs on the plate. Over the course of an hour, plates were closely monitored for hatching and newly hatched young L1 animals were subjected to assays.

### Assays

To obtain locomotion data across the stages of development, worms synchronized at each stage were subjected to swimming and crawling assays in which their movements were captured on video. All assays were conducted on NGM plates with no bacterial lawn. For crawling assays, worms were starved for one hour prior to the start of the assay. A platinum wire worm pick was used with Halocarbon 700 oil to transfer 20 worms onto a 90mm NGM plate in the absence of OP50. We took 1–2 minute long video clips of each crawling worm. For swimming assays, each worm was transferred using a pick into a 5, 10, or 15μl drop of M9 buffer solution placed on the surface of the assay plate for L1-L2, L3, and L4-adult animals, respectively. M9 droplets were flattened with a worm pick in a circular motion beforehand to reduce glare. For each assay, one minute was allowed for the worm to acclimate to swimming conditions before 1–2 minutes of the worm's locomotion was tracked and analyzed. For gait transition assays we followed the crawling protocol for adult animals, however, we added a 1.5μl drop of M9 buffer to the plate in the worm's path following previously reported protocols [44].

### Tracking

WormLab imaging stations were used in conjunction with WormLab software (both from MBF Bioscience) to capture videos of the worms and subsequently track the curvature data of each worm. Additionally, for young and late L1 larval assays, a macro lens (LAOWA 25mm F2.8 2.5-5x ULTRA MACRO) at 2.5x magnification was mounted instead of the default lens in order to capture the small worms at high resolution. All videos were taken at 1200x1600 resolution and 14 frames per second. To ensure worms remained in the field of view, assay plates were gently moved if necessary. For each worm, the angles between each of the eleven segments along the length of the animal (as defined by the WormLab software) were used to define the worm's curvature for each video frame. We segmented the animal into 5, 9, 11, 17, and 33 segments and found no significant differences in eigenworm shape analyses with more than 11 segments on a small held-out test data set. Subsequently, we proceeded with 11 segments (10 subsequent segment angles) for the remainder of the study.

## Data

All worm segment curvature data is publicly available at https://doi.org/10.5061/dryad.
stqjq2c8p [87]. TXT files are available for every worm assayed and tracked in these studies.
Files are organized into subfolders by locomotion gait and developmental age. Each TXT file
has 11 columns, the first is for time stamps, and the remaining for the 10 segment angles
(radians).

## Eigenworm analysis

We denote the worm's posture as $\theta(s)$ where s denotes the segment number. We perform an
eigen-decomposition by first constructing the covariance matrix of the postures as:
$C(s, s\prime) = \langle (\theta(s) - \langle \theta \rangle)(\theta(s\prime) - \langle \theta \rangle) \rangle$

Eigenworms $\mu(s)$ and eigenvalues $\lambda_i$ are defined by the eigendecomposition:
$\sum_{s'} C(s, s')\mu_i(s') = \lambda_i \mu_i(s)$

The cumulative variance explained is defined by:

$$\sigma_k^2 = \sum_{i=1}^{k} \frac{\lambda_i}{\sigma^2}$$

For all eigenworm amplitude figures, we define the eigenworms from the covariance matrix
of posture angles across all developmental stages for that gait type. This choice is justified as
the covariance matrix calculated from each separate developmental stage has nearly identical
eigenvectors (Figs 3A and 4A), meaning there is no loss of descriptive power by combining
stages into a single covariance matrix. This combination allows for comparisons across all
developmental stages from the same perspective.

## Kernel density estimate

We visualize the eigenworm amplitude distributions through a kernel density estimate of the
eigenworm amplitudes. A kernel density estimate is a smoothed histogram; data are binned in
two dimensions and then smoothed using a kernel. We used a gaussian kernel with a band-
width determined by Scott's rule, implemented by the Python function *scipy.stats.
gaussian_kde*.

## Speed analysis

We calculate speed at each frame by calculating the Euclidean distance between subsequent
points in eigenworm space and multiplying them by the sampling frequency. We infer the
phase in each frame by taking the Hilbert transform of the first eigenworm amplitude.

## Participation Ratio

We utilize a continuous measure of dimensionality derived from the eigenvalues of the posture
covariance matrix called the participation ratio (PR). The PR can be thought of as the dimen-
sions required to capture approximately 80% to 90% of the variance of the data [54]. The par-
ticipation ratio is defined as:

$$PR = \frac{\left(\sum_i \lambda_i\right)^2}{\sum_i \lambda_i^2}$$

In the simple case of three-dimensional (N = 3) data, if the eigenvalues are 1, 0, 0, the PR is
1. If that same data were evenly distributed with eigenvalues of ⅓, ⅓, ⅓, it would have a PR of

3. Most data will contain some correlational structure that will place the PR somewhere in between the values of 1 and N, with N being the number of features in the data.

We calculated the participation ratio of each recording individually and collected the distribution of participation ratios for each age group. We performed a standard independent two sample t-test assuming equal sample variance using the scipy ttest_ind function in Python. We corrected alpha values for multiple comparisons using Bonferroni adjustment yielding a significant p-value of 0.05/15 or 0.0033.

## Code

All analyses of worm segment data were performed using custom Python scripts relying primarily on the *matplotlib*, *numpy*, *scipy* libraries. Code to reproduce figures as well as instructive example notebooks are hosted at https://github.com/ssterrett/wopodyn.

## Supporting information

**S1 Fig. Adult *C. elegans* swimming and crawling are distinct in eigenworm space.** (a and b) Scatter plots of swimming and crawling data from a swimming to crawling transitions assay in the plane of (a) eigenworm 1 and eigenworm 2 and (b) eigenworm 3 and eigenworm 4. (EPS)

**S1 File. Adult *C. elegans* rhythmic swimming video.** Representative 10 second video of adult *C. elegans* swimming aligned with kymograph and eigenworm amplitude probability density estimates. (MOV)

**S2 File. Adult *C. elegans* rhythmic crawling video.** Representative 10 second video of adult *C. elegans* crawling aligned with kymograph and eigenworm amplitude probability density estimates. (MOV)

**S3 File. Young L1 *C. elegans* disrupted swimming video.** Representative 10 second video of young L1 *C. elegans* swimming aligned with kymograph and eigenworm amplitude probability density estimates. (MOV)

**S4 File. Young L1 *C. elegans* disrupted crawling video.** Representative 10 second video of young L1 *C. elegans* crawling aligned with kymograph and eigenworm amplitude probability density estimates. (MOV)

**S1 Table. Table of definitions.** (PDF)

## Acknowledgments

We thank the Caenorhabditis Genetics Center for providing strains. We would like to thank multiple mentorship programs through the Fred Hutch Cancer Center, SURP Undergraduate Researchers and Pathways Undergraduate Researchers, and through Pomona College Internship Program for facilitating relationships with undergraduate trainees who helped complete this project. We thank Kailee Padron for help with synchronization of worms. We thank Lin Zhang for discussions on locomotor assay protocols. The authors thank Monet Jimenez and Irini Topalidou, for critical reading of the manuscript.

## Author Contributions

**Conceptualization:** Cera W. Hassinan, Scott C. Sterrett, Jihong Bai.

**Data curation:** Cera W. Hassinan, Brennan Summy, Angie Wang.

**Formal analysis:** Scott C. Sterrett, Arnav Khera.

**Funding acquisition:** Jihong Bai.

**Investigation:** Cera W. Hassinan, Brennan Summy, Angie Wang.

**Methodology:** Cera W. Hassinan, Scott C. Sterrett, Brennan Summy, Jihong Bai.

**Project administration:** Cera W. Hassinan, Jihong Bai.

**Resources:** Jihong Bai.

**Software:** Scott C. Sterrett, Arnav Khera.

**Supervision:** Cera W. Hassinan, Scott C. Sterrett, Jihong Bai.

**Validation:** Cera W. Hassinan, Scott C. Sterrett, Brennan Summy, Jihong Bai.

**Visualization:** Cera W. Hassinan, Scott C. Sterrett.

**Writing – original draft:** Cera W. Hassinan, Scott C. Sterrett, Brennan Summy.

**Writing – review & editing:** Cera W. Hassinan, Scott C. Sterrett, Jihong Bai.

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
