## [Decision Letter · Decision Letter 0]

8 Jan 2024

Dear Dr Bai,

Thank you very much for submitting your manuscript "Dimensionality of locomotor behaviors in developing C. elegans" for consideration at PLOS Computational Biology. As with all papers reviewed by the journal, your manuscript was reviewed by members of the editorial board and by several independent reviewers. The reviewers appreciated the attention to an important topic. Based on the reviews, we are likely to accept this manuscript for publication, providing that you modify the manuscript according to the review recommendations.

As you will see from the comments, both reviewers were very positive about the work, but one makes several suggestions that should improve the clarity of the paper.

Sincerely,

Barbara Webb

Academic Editor

PLOS Computational Biology

Natalia Komarova

Section Editor

PLOS Computational Biology

Reviewer's Responses to Questions

**Comments to the Authors:**

Reviewer #1: This study focuses on the evolutionary trajectory of locomotion throughout development. Employing principal component analysis, the authors found a consistent, low number of eigenworms in both larvae and adult worms during swimming and crawling movements. However, a noteworthy observation is the increased coordination in locomotion as development progresses. Significantly, this research bridges a substantial gap in our comprehension of locomotor behaviours during the developmental stages of C. elegans, representing a valuable addition to the existing knowledge in this field.

The experimental design is commendable, with comprehensive data analysis leading to robust results that strongly support the authors' conclusions. This work not only enhances our understanding of locomotor development and gait transitions in C. elegans but also establishes a crucial framework for future investigations into the modulatory mechanisms of the establishment, maintenance, and adaptability of locomotion during development. I strongly recommend the acceptance of this manuscript due to its significant contributions to the field.

Reviewer #2: This paper investigates how C. elegans crawl and swim as they grow from young L1 larvae to adulthood. They used PCA to show that ~97% of the variance in crawling and swimming behaviors can be captured by the first 4 eigenworms and that these eigenworms are different for crawling and swimming locomotion. They then found that young L1 larvae are able to form the eigenworm shapes for swimming and crawling but they don't transition in the robust cyclical way that adults do and are therefore unable to locomote robustly until the late L1 stage when they do show robust rotations in eigenworm space.

This paper is well written and easy to understand and illustrates some interesting features in developing C. elegans as they learn to move around their environments, and the scientific method is sound. I have a few minor changes to suggest and questions below.

The table of definitions is unnecessary as these terms are mostly described already in the text. Perhaps put it in the supplement if anything.

Is there a reason 11 segments is the right number or was it just used for convenience to have 10 angles to look at? Has there been any work done to optimize the number of segments to maximize the variation captured by the fewest eigenworms or anything along those lines?

Line 177 - Does the constant density in eigenworm space show that it's a constant propagating wave along the worm? I think it shows that the worm goes constantly between eigenworm one and eigenworm two states which seem to have slightly different wavelengths, ~1.5 waves per worm and ~2 waves per worm. From the videos of course it looks like a constant propagating wave I'm just not sure that is what this plot directly shows and there may be a step missing in the logic here that could be elaborated on.

Line 183 / Figure 2g - You say that the first four eigenworms in the transition assay match the first two for each crawling and swimming locomotion. It strikes me that plotting the crawling and swimming rings in the space of the first three eigenworms is missing out on one of the eigenworms for one type of locomotion. Instead perhaps you can make two plots with these two rings on them one for the crawling eigenworms and one for the swimming ones? For example if eigenworms one and three are the same as the crawling eigenworms and two and four are those of swimming can you make two 2D plots with both swimming and crawling locomotion rings on them. One plot that is eigenworm one vs. eigenworm three and one that is eigenworm two vs. eigenworm four. Then I would expect that the orange crawling points would trace out a nice circle in the eigenworm one vs. eigenworm three plot while they blue points would be more scattered (?) in this space, and vice versa.

Line 218 - I am slightly confused by the description of PR. Figure 1d seems to show that 85% of the variation is captured by the first two eigenworms, wouldn't that make the PR for adult worms 2 or less since it is already above 80% there?

~Line 235 - Do the worms grow uniformly along their length? Since the ancestry of all the cells in adult worms is known can you segment the different growth stages at the same cell rather than at equally spaced points?

**Have the authors made all data and (if applicable) computational code underlying the findings in their manuscript fully available?**

Reviewer #1: Yes

Reviewer #2: Yes

PLOS authors have the option to publish the peer review history of their article (what does this mean?). If published, this will include your full peer review and any attached files.

Reviewer #1: No

Reviewer #2: **Yes: **Katherine Copenhagen

Figure Files:

Data Requirements:

Reproducibility:

References:

---

## [Decision Letter · Decision Letter 1]

12 Feb 2024

Dear Dr Bai,

We are pleased to inform you that your manuscript 'Dimensionality of locomotor behaviors in developing *C. elegans*' has been provisionally accepted for publication in PLOS Computational Biology.

Best regards,

Barbara Webb

Academic Editor

PLOS Computational Biology

Natalia Komarova

Section Editor

PLOS Computational Biology

Reviewer's Responses to Questions

**Comments to the Authors:**

Reviewer #2: The updates to the manuscript addressed all of the comments and questions that I had.

**Have the authors made all data and (if applicable) computational code underlying the findings in their manuscript fully available?**

Reviewer #2: Yes

PLOS authors have the option to publish the peer review history of their article (what does this mean?). If published, this will include your full peer review and any attached files.

Reviewer #2: **Yes: **Katherine Copenhagen

---

## [Editor Report · Acceptance letter]

26 Feb 2024

PCOMPBIOL-D-23-01560R1 

Dimensionality of locomotor behaviors in developing C. elegans

Dear Dr Bai,

I am pleased to inform you that your manuscript has been formally accepted for publication in PLOS Computational Biology. Your manuscript is now with our production department and you will be notified of the publication date in due course.

With kind regards,

Anita Estes
